# Amount, Preparation and Type of Formula Consumed and Its Association with Weight Gain in Infants Participating in the WIC Program in Hawaii and Puerto Rico

**DOI:** 10.3390/nu11030695

**Published:** 2019-03-24

**Authors:** Rafael E. Graulau, Jinan Banna, Maribel Campos, Cheryl L. K. Gibby, Cristina Palacios

**Affiliations:** 1Nutrition Program, Graduate School of Public Health, Medical Sciences Campus, University of Puerto Rico, P.O. Box 365067, San Juan, PR 00936-5067, Puerto Rico; rafael.graulau@upr.edu; 2Department of Human Nutrition, Food and Animal Sciences, College of Tropical Agriculture and Human Resources, University of Hawaii at Manoa, Agricultural Sciences 216, 1955 East-West Rd, Honolulu, HI 96822, USA; jcbanna@hawaii.edu (J.B.); clkkoide@hawaii.edu (C.L.K.G.); 3Dental and Craniofacial Genomics Core, Endocrinology Section School of Medicine, University of Puerto Rico, P.O. Box 365067, San Juan, PR 00936-5067, Puerto Rico; maribel.campos@upr.edu; 4Department of Dietetics and Nutrition, Robert Stempel College of Public Health & Social Work, Florida International University, 11200 SW 8th Street, AHC 5-313, Miami, FL 33199, USA

**Keywords:** infant, feeding, formula, over-feeding, Hispanics, Native Hawaiians

## Abstract

The aim of this study was to assess the association between amount (below or above recommendations), preparation (liquid vs. powder), and type (regular vs. hydrolysate) of infant formula consumed and weight in infants participating in the Women, Infant and Children (WIC) Program in Hawaii (HI) and Puerto Rico (PR). This was a secondary analysis of 162 caregivers with healthy term 0–2-month-old infants. Socio-demographics, infant food frequency questionnaires, and weight and length were assessed at baseline and after four months. Infant feeding practices were associated with weight-for-length z-scores using multivariable logistic regression. In total, 37.7% were exclusively breastfed and 27.2% were exclusively formula-fed. Among formula users, regular (63.6%) and powder (87.0%) formula were the most common; 43.2% consumed formula above recommendations. Most infants had rapid weight gain (61.1%). Infants fed regular formula had higher odds of overweight after four months (adjusted OR = 8.77, 95% CI: 1.81–42.6) and higher odds of rapid weight gain (adjusted OR = 3.10, 95% CI: 1.12, 8.61). Those exclusively formula fed had higher odds of slow weight gain (adjusted OR = 4.07, 95% CI: 1.17–14.2). Formula preparation and amount of formula were not associated with weight. These results could inform the WIC program’s nutrition education messages on infant feeding. Studies with longer follow-up are needed to confirm these results.

## 1. Introduction

Prevalence of infant obesity in the US has been estimated at 8.1% [1]. This is higher among infants participating in the Special Supplemental Nutrition Program for Women, Infants, and Children (WIC) (12.3%) [2], particularly among Hispanic (16.4%) and Native Hawaiian (19.3%) compared to African American (13.0%) and White (13.9%) infants [2].

Early life growth patterns and behaviors have been recognized as playing an important role in obesity, even though these might not be evident during the early stages of life [3]. Growth is more rapid during early infancy than at any other time during the human life cycle, and studies have confirmed the association between rapid weight gain and risk of obesity later in life [4,5,6,7], particularly if the rapid weight gain occurs within the first months of life [8]. Therefore, early modifiable risk factors could have an impactful influence on the long-term health of infants.

Multiple factors affect the growth trajectory in infant, but nutrition is considered a key determinant [9]. Breastfeeding has been long recognized to help reduce the risk of obesity compared to non-breastfeeding [10,11]. A physiological reason for this is that the protein intake of breast-fed infants decreases with age and resembles the protein requirements during the early months of life, while the protein content of standard formulas surpass these requirements after the first couple of months [11]. On the contrary, formula feeding has been associated with a higher risk of obesity. An analysis of 70,000 infants enrolled in the WIC Program found that obesity at age 4 was higher in infants fed only formula (25.9%) compared to those exclusively breastfed (19.8%) [12]. Additionally, very recent longitudinal studies have shown that infants fed formula during the first 3 months of life gain significantly more weight compared to breastfed infants at six or nine months of age [13,14,15]. Formula overfeeding (feeding above recommended guidelines by age) appears to also influence weight, since bottle-fed infants have less control over feeding volumes [16].

Breastfeeding is the feeding method recommended by the World Health Organization [17]. However, breastfeeding rates are low in the US. Data from the WIC program, a nutrition supplementation program in the US with more than 1.93 million infants participating nationally in 2016, showed that only 30.9% were breastfed (12.9% exclusively breastfed and 18.0% partially breastfed) while 69.1% were formula fed [18]. The most common type of formula used in the US as recently reported from infant formula sales are regular casein or whey based (not hydrolysate) (63%), followed by hydrolysate formula (partially or extensively hydrolysate) [19].

However, very little is known about the association between use of different types of formula, amounts or preparation methods and weight gain early in life. Hydrolyzed formulas appeared to be more easily absorbed, may exert different satiety responses and may also affect weight differently compared to regular formulas [20]. Additionally, consumption of formula above recommended guidelines may also lead to rapid weight gain [21]. Furthermore, reconstitution errors when preparing powder formulas could lead to overfeeding and thus rapid weight gain [22].

The present study evaluated the association between the type of feeding, formula amount, formula type, and formula preparation during the first two months of life with the risk of overweight and rapid weight gain four months later among infants participating in the WIC program in Hawaii and Puerto Rico. Results from this study could contribute to the new Dietary Guidelines for Americans being developed for children from birth to two years of age in the US [23]. Although breastfeeding is the recommended method, this is not the main feeding method used in the US and studies are needed to understand the contribution of formula feeding to obesity.

## 2. Methods

### 2.1. Study Population

This is a secondary analysis of data collected in participants recruited for a multi-site trial using short mobile messages (SMS) to improve infant weight in low-income minorities conducted in WIC clinics in PR and HI [24,25]. A total of 202 parents/caregivers of healthy term infants aged 0–2 months old participating in the WIC program were recruited from two WIC clinics in PR (100 participants) and four WIC clinics in HI (102 participants). Participants were recruited from the waiting areas of the WIC clinics by the research team. Those interested in the study were screened for eligibility criteria which included: caregiver age 18 years and older, owner of a mobile phone, responsible for infant care, infants without special diets or limited mobility, healthy term birth (≥37 weeks), adequate birthweight (≥10th or ≤90th percentile), and being able to read. A convenience sample was recruited from January to May of 2016, which was based on availability of resources. All study visits were conducted at the WIC clinics. Details of the main study are published elsewhere [24,25].

For the present analysis, data from the infant food frequency questionnaire (FFQ) collected at the initial visit (when infants were 0–2 months) and anthropometric measures collected at the initial visit and at the follow-up visit (four months later) were analyzed to quantify the amount and type of formula consumed in infants 0–2 months and examine its association with weight status and weight gain in the next four months. A total of 40 participants were excluded due to incomplete anthropometric measurements and/or feeding practices information. Consequently, a total of 162 participants were included in the analysis.

The main study was approved by the Institutional Review Board of the University of Puerto Rico, Medical Sciences Campus and the University of Hawaii at Manoa, in collaboration with the WIC Program at both sites. Participants provided written informed consent before the study began.

#### 2.1.1. Socio-Demographic Characteristics

Participants completed a short socio-demographic questionnaire at the initial visit. It included questions about caregivers’ age, race/ethnicity, level of education, number of children, infants’ age, sex, weight and length at birth. It also included a question about weight gain during pregnancy; this was classified as low, adequate, or high based on the recommended weight gain by the Institute of Medicine [26].

#### 2.1.2. Infant Food Frequency Questionnaire (FFQ)

We used a FFQ to collect information on type, amount and preparation methods of infant formulas. For this, participants completed a validated infant FFQ [27]. Briefly, this semi-quantitative FFQ includes a total of 52 foods commonly consumed by toddlers with a short description of its source and/or preparation. The frequency of consumption of each food was assessed as feedings per day for foods consumed daily or per week for foods consumed less often; for the latter, the frequency was divided by 7 in order to obtain the total amount of each food item per day. The frequency was then multiplied by the serving size. The FFQ was administered through a face-to-face interview with trained research personnel. Photographs from a booklet with baby food items were shown to aid participants in providing precise estimates of portion sizes. From this FFQ, we focused on type of milk consumed (i.e., exclusive breastfeeding, both breastmilk and formula feeding, and exclusive formula feeding), type of formula consumed (i.e., not hydrolysate, partially/extensively hydrolysate, and other), and preparation of formula (i.e., powdered, concentrated, or ready-to-use). The amount of formula was calculated taking into account the amount of formula consumed at each feeding (in ounces) and the frequency reported per day. It was then classified as: (1) low, if consumption was less than 16 ounces/day, (2) adequate, if consumption was between 16–24 ounces/day, or (3) rapid, if consumption was more than 24 ounces/day, based on the WIC and the Food and Nutrition Services guidelines for infants this age [28,29]. To evaluate formula feeding as a risk factor for rapid weight gain, infants were re-classified as: (1) any breastfeeding or (2) only formula feeding.

#### 2.1.3. Infant’s Weight and Length

The main outcome variables in these analyses were weight status at the follow-up visit and weight change between the initial (when infants were 0–2 months old) and follow-up (four months later) visits. Infants were weighed and measured by trained research personnel at each time point at the WIC clinics. Length was assessed in duplicates using the infant WIC stadiometer. Weight was also obtained in duplicates using the infant WIC scale with light clothes, no shoes, and clean diaper. The average of the two measurements was used in the analysis. Weight-for-length (WHL) z-scores were calculated at each time point using the WHO AnthroPlus macro [30] and compared using the World Health Organization Growth Charts [31,32]. A z-score < −2 was considered underweight, a z-score > −2 to < 2 was considered adequate weight, while a z-score > 2 was considered obese [31]. Rapid weight gain was defined as a change greater than 0.67 standard deviations (SD) in weight-for-length z-score between the first and second assessment [4]. Adequate weight gain was considered as a change in weight-for-length z-score between −0.67 to +0.67 SD, while a slow weight gain was that below −0.67 SD.

#### 2.1.4. Statistical Analyses

Measures of frequency distributions were performed to describe categorical variables and summary measures to describe continuous variables. We did not conduct a power analysis; therefore, we are reporting only effect sizes. A multivariable logistic regression model was used to evaluate the different feeding variables potentially associated with rapid weight gain and weight status at the follow-up period, such as feeding method, formula overfeeding, type of formula consumed, and preparation of formula. Bivariate analyses were conducted between the main outcome variables and socio-demographic variables to identify confounders for which to adjust in the regression model. We also included other confounders based on the literature. Results were expressed as the odds ratio (95% confidence interval) and adjusted odds ratio (95% confidence interval) of weight status at the follow-up period and weight change from the initial to the follow-up period for each feeding variable in relation to the reference category. We also planned to stratified results by initial weight status category. Data were analyzed using SPSS (version 17.0, SPSS, Inc., Chicago, IL, USA).

## 3. Results

Socio-demographic characteristics of caregivers and infants are shown in Table 1. The median age of caregivers was 27.0 years and the median number of children was two. Most caregivers had a level of education of no more than a high school diploma (67.9%), 45.1% had an adequate pre-pregnancy BMI, and 43.2% had an adequate pregnancy weight gain. The median age of infants at baseline was 1.00 month, with equal sex distribution, and adequate birth weight (median 3.30 kg).

Table 2 shows infant feeding practices at the initial visit. A total of 37.7% of infants were exclusively breastfed, 35.2% were both breast and formula fed, and 27.2% were exclusively formula fed. Among those consuming formula, regular formula (not hydroxylate) was more common (63.6%). Most caregivers used powdered formula (87.0%) and 43.2% consumed formula above current recommendations.

Table 3 shows that most infants had adequate weight status at the initial visit (86.6%) and at the follow-up visit (70.5%). However, most had rapid weight gain during this period (61.1%). When stratified by weight status at the initial visit, we found that most underweight infants had rapid weight change during this period, while most of those categorized as overweight in the initial visit had slow weight change during this period.

Table 4 shows the associations between the feeding practices and weight status at the follow-up period. We found that infants fed regular formula (not hydrolysate) had 8.77 higher odds of being overweight at the follow-up visit compared to those fed hydrolysate formula, even after adjusting for important confounders (adjusted OR = 8.77, 95% CI: 1.81–42.6). Type of feeding (breastfeeding vs. formula feeding), formula type preparation (liquid vs. powder) and amount of formula (below or above recommendations) were not associated with weight status at the follow-up visit.

Table 5 shows the associations between the feeding practices and weight gain between the initial and final visit. We found a 4.07 higher odds of slow weight gain in infants exclusively formula fed compared to those breastfed (adjusted OR = 4.07, 95% CI: 1.17–14.2) but no significant association between type of feeding and rapid weight gain. Compared to hydrolysate formula use (partially or extensive), using regular formula (not hydrolysate) was associated with a three-fold increase of rapid weight gain in both the un-adjusted model (OR = 3.39, 95% CI: 1.25, 9.16) and in the adjusted model (OR = 3.10, 95% CI: 1.12, 8.61). Formula type preparation (liquid vs. powder) and amount of formula (below or above recommendations) were not associated with slow or rapid weight gain. We could not run the logistic regressions stratified by weight status at the initial visit due to low number of underweight and overweight infants per feeding categories.

## 4. Discussion

The present study among healthy term infants participating in the WIC program in Hawaii and Puerto Rico found that infants fed regular formulas (not hydrolysate) had higher odds of being overweight at the follow-up visit and of having rapid weight gain during the four months of follow-up compared to those fed hydrolysate formula. We also found higher odds of slow weight gain in infants exclusively formula fed compared to those breastfed. Formula preparation (liquid vs. powder) and amount of formula (below or above recommendations) were not associated with weight status at the follow-up visit or with slow or rapid weight gain.

This is the first study to show that formula intake increases the odds of slow weight gain compared to breastfeeding, although it was not associated with weight status at four months. Most prospective studies have found a positive association between formula feeding and rapid weight gain [13,14,15,33,34,35,36,37], showing a different pattern of weight gain by type of feeding. For example, the Darling study following 87 infants (46 breastfed and 41 formula fed) for 12 months showed similar WHL z-scores between breastfed and formula fed infants during the first four months, but then from the 5th to 18th month, WHL z-score was consistently higher among formula fed infants [35]. The Canadian Healthy Infant Longitudinal Development (CHILD) birth cohort, which followed 2553 infants from birth up to 12 months of age, also found that non-exclusive or partial breastfeeding for 3 or 6 months was associated with rapid weight gain [37]. The potential protective effects of exclusive breastfeeding on weight gain may be through a modulation of growth rate during the first year. This may explain why in our study we did not see this potential protective effect of breastfeeding, as we only followed infants for four months. Additionally, it is important to note that most studies focus on rapid weight gain and not slow weight gain. One of the few studies to evaluate this was an analysis from the Avon longitudinal study in the UK; this analysis found that mothers with infants with slow weight gain were more likely to stop breastfeeding by week 4 after birth [38]. In the present analysis, the proportion of infants that had stopped breastfeeding by four weeks was similar among those with adequate, rapid or slow weight gain. We did see, however, that most infants with slow weight gain were overweight at baseline, and caregivers may have been responding to their perception of their infants’ weight. This has to be evaluated further.

Furthermore, breastfeeding may help in the self-regulation of the amount of milk consumed [39]. It is still not exactly clear the exact mechanism, but several proteins present in milk have been proposed as possible modulators, such as leptin [40], adiponectin [41], and IGF-1 [42]. Additionally, the protein content in breastmilk may also be involved [43]. Although formula has somewhat different proteins than breastfeeding, the type of protein may explain the results found by type of formula in the present study. Specifically, we found that infants fed regular formulas (not hydrolysate) had higher odds of overweight at the follow up visit and of rapid weight gain during the study compared to infants fed hydrolysate formulas. There are very limited studies evaluating the relationship between type of formula and weight gain. A randomized, double-blind study among 205 infants testing regular formula, partially hydrolyzed formula, and human milk for four months starting at birth found similar weight gain among all groups [44]. A more recent study evaluating the long-term effects (three and five years of age) of consuming regular versus extensively or partially hydrolyzed formulas during the first year of life showed no differences in growth outcomes [45]. However, two clinical trials conducted by Mennella’s group among infants comparing regular versus hydrolyzed formulas for several months found that those consuming the hydrolyzed formula had a significantly lower weight gain velocity and consumed less to reach satiation than those on the regular formula [46,47,48]. As mentioned earlier, protein in hydrolysate formulas is hydrolyzed and appears to be more easily absorbed; as shown in the aforementioned study, it may exert different satiety responses and, therefore, affect weight gain [20]. More studies are needed to understand the long-term effects of consuming each type of formula on rapid weight gain in young children.

Formula preparation (liquid vs. powder) was not associated with weight. There are very few studies evaluating this. A study involving 43 newborns found that infants fed powder formula had significantly greater weight gain than those using ready-to-feed (liquid) formula when they were three- and six-months old [22]. The potential association of powdered formula with weight gain could be explained by errors in reconstituting milk, as parents could be preparing more concentrated formulas so that infants sleep longer or for other reasons, but this has to be evaluated in more detail. In the present study, we did not observe errors in the reconstitution of formula (results not shown). Similar, amount of formula (below or above recommendations) was also not associated with weight in the present study, contrary to a few other reports. For example, in another cross-sectional study conducted by our group among 296 infants and toddlers aged 0–24 months of age participants of the WIC Program in Puerto Rico, we found that formula overfeeding was associated with greater odds of obesity, even after adjusting for sex and age of the baby and education of the parents (OR = 2.52; 95% CI: 1.03, 6.15) [21]. Another study that assessed the size of the bottle used for feeding found that caregivers who used larger feeding bottles (≥6 oz) gave four more ounces of formula per day, and this was significantly associated with rapid weight gain [9,49]. In addition, results from the Avon Longitudinal Study of Parents and Children in UK, which followed 1112 infants from birth until 10 years of age, found that children who were fed ≥600 mL (≥21 oz) per day had greater weight gain and also higher BMI, which persisted until later in childhood [50].

It is interesting to note that infants who were categorized as underweight in the initial visit all had rapid weight gain during the study period. On the contrary, most of the infants categorized as overweight initially had slow weight gain. The concept of catch-up is usually related to height velocity for infants with periods of growth inhibition [51] or with catch-up fat or weight among infants born small for gestational age [52]. However, in the present study among a sample of healthy term infants with adequate birthweight, we did see a catch-up in weight gain in only four months of follow-up. A longer follow-up is needed to understand how weight gain is modified beyond this period.

There are some limitations of the present study. This was a secondary analysis and as such, it was not designed to answer this question directly. The alternative type of formula to cow’s milk formula that was evaluated in the study can be partially or extensively hydrolyzed, and this was not taken into account because participants did not specify. However, the WIC program does not include in their regular food package extensively hydrolyzed formulas; therefore, most formulas used were probably partially hydrolyzed. The final sample size was small and infants were only followed for four months; larger sample size with longer follow-up may be needed. Furthermore, all feeding practices were self-reported at baseline. However, all anthropometric measurements were performed using standardized methods in both sites. In addition, there are very limited longitudinal data relating infant feeding practices with weight gain, particularly among high-risk minority groups. Therefore, a longitudinal study following infants from birth until later in childhood taking into account different feeding practices and other practices such as sleep and sedentary behaviors is needed.

## 5. Conclusions

In conclusion, type of feeding (formula vs. breastfeeding) and type of formula (regular vs. hydrolysate) were associated with weight gain. However, consumption of formula below or above recommendations and preparation type (powdered vs. liquid formulas) were not. These results could inform the WIC program’s nutrition education messages on type of feeding and type of formulas for promoting adequate weight gain. However, studies with longer follow-up are needed to confirm these results.

## Figures and Tables

**Table 1 nutrients-11-00695-t001:** General characteristics of caregivers and their infants at baseline (*N* = 162).

Variables	%(*n*) or Median (25th, 75th percentiles)
*Caregivers*	
**Mother’s Age (years)**	27.0 (23.0, 31.0)
**Ethnicity**	
Hispanics	63.0 (102)
Non-Hispanics	37.0 (60)
**Race**	
White	45.1 (73)
Black	15.4 (25)
Asian, Native Hawaiian, Pacific	39.5 (64)
**Education**	
≤High school	67.9 (110)
≥College degree	32.1 (52)
**Number of children**	2.00 (1.00, 2.00)
**Pre-pregnancy BMI**	24.8 (21.6, 30.7)
Underweight	6.00 (10)
Normal	45.1 (73)
Overweight	23.5 (38)
Obese	25.3 (41)
**Pregnancy weight gain (lbs)**	26.0 (20.0, 35.0)
Low	21.6 (35)
Adequate	43.2 (70)
Excessive	35.2 (57)
***Infants/toddlers***	
**Age (months)**	1.00 (0.68, 1.35)
**Sex**	
Male	50.0 (81)
Female	50.0 (81)
**Birth weight (kg)**	3.30 (2.95, 3.57)

**Table 2 nutrients-11-00695-t002:** Feeding patterns at baseline (*N* = 162).

Variables	% (*n*) or Median (25th, 75th percentiles)
**1. Type of feeding**	
Exclusive breastfeeding	37.7 (61)
Partial breastfeeding	35.2 (57)
Exclusive formula feeding	27.2 (44)
**2. Type of formula ^1^**	
Not hydrolysate	63.6 (63.6)
Partially/extensively hydrolysate	35.4 (35.4)
Soy-based	1 (1.0)
**3. Formula preparation ^1^**	
Liquid-ready to use	11.0 (11)
Liquid-concentrated	2.0 (2)
Powder	87.0 (87)
**4. Amount of formula (oz) ^2^**	24 (18.5, 32.0)
Below recommendations	9.1 (4)
Adequate	47.7 (21)
Above recommendations	43.2 (19)

^1^ Among those that consumed any amount of formula; missing one participant that did not report type, preparation, or amount of formula. ^2^ Among those that consumed only formula.

**Table 3 nutrients-11-00695-t003:** Weight status in infants during the study (*N* = 162).

**Weight Status**	**Initial Visit** **(0–2 months)**		**Follow-Up Visit** **(4–6 months)**	
	**% (*n*)**		**% (*n*)**	
**Weight-for-length z score**				
Underweight (<−2 z score)	7.4 (12)		1.2 (2)	
Adequate (−2 to 2 z score)	86.4 (140)		72.8 (118)	
Overweight/obese (>2 z score)	6.2 (10)		25.9 (42)	
**Change in weight-for-length z score^1^**	**Overall**	**By weight status at initial visit**		
		**Underweight**	**Adequate**	**Overweight**
Slow weight change (<−0.67 SD)	13.0 (21)	0	10.7 (15)	60.0 (6)
Adequate weight gain (−0.67 to 0.67 SD)	27.8 (45)	0	30.7 (43)	20.0 (2)
Rapid weight change (>0.67 SD)	59.3 (96)	100 (12)	58.6 (82)	20.0 (2)

^1^ Only two infants were categorized as underweight at the follow-up visit, therefore, they were excluded from the analysis.

**Table 4 nutrients-11-00695-t004:** Factors associated with overweight/obesity at the follow-up visit (*N* = 162) ^1^.

Feeding Variables	Overweight/Obesity at the Follow-Up Visit			
	OR (95% CI)	*p* Value	Adjusted OR (95% CI) *	*p* Value
**1. Type of feeding**				
Any breastfeeding	1		1	
Only formula	0.98 (0.44, 2.19)	0.969	0.74 (0.31, 1.80)	0.510
**2. Type of formula ^1^**				
Hydrolysate (partially or extensive)	1		1	
Not hydrolysate	10.86 (2.39-49.3)	0.002	8.77 (1.81-42.6)	0.007
**3. Formula preparation**				
Liquid	1		1	
Powder	1.88 (0.38, 9.30)	0.442	1.50 (0.28, 10.5)	0.680
**4. Amount of formula**				
Adequate	1		1	
Below recommendations	0.67 (0.32, 1.43)	0.307	0.75 (0.33, 1.71)	0.494
Above recommendations	0.68 (0.20, 2.36)	0.544	0.72 (0.18, 2.87)	0.637

^1^ Data from one infant consuming soy formula and two underweight infants at the follow-up visit were not included in the analysis. *Adjusted for age of the mother and infant, sex of the infant, mother’s level of education, and pregnancy weight gain status.

**Table 5 nutrients-11-00695-t005:** Factors associated with weight gain during the follow-up period (*N* = 162).

Feeding Variables	Slow Weight Gain during the Follow-Up Period				Rapid Weight Gain during the Follow-Up Period			
	OR (95% CI)	*p* Value	Adjusted OR (95% CI) *	*p* Value	OR (95% CI)	*p* Value	Adjusted OR (95% CI) *	*p* Value
**1. Type of feeding**								
Any breastfeeding	1		1		1		1	
Only formula	3.62 (1.09, 12.07)	0.036	4.07 (1.17, 14.2)	0.028	2.19 (0.87, 5.50)	0.096	2.04 (0.80, 5.22)	0.137
**2. Type of formula ^1^**								
Hydrolysate (partially or extensive)	1		1		1		1	
Not hydrolysate	1.52 (0.43, 5.43)	0.519	1.64 (0.44, 6.17)	0.512	3.39 (1.25, 9.16)	0.016	3.10 (1.12, 8.61)	0.030
**3. Formula preparation**								
Liquid	1		1		1		1	
Powder	0.33 (0.05, 2.30)	0.265	0.15 (0.02, 1.52)	0.108	0.79 (0.15, 4.21)	0.780	0.54 (0.09, 3.40)	0.509
**4. Amount of formula**								
Adequate	1		1		1		1	
Below recommendations	0.58 (0.19, 1.84)	0.583	0.61 (0.19, 2.06)	0.446	0.60 (2.8, 1.30)	0.598	0.58 (0.26, 1.29)	0.182
Above recommendations	1.31 (0.23, 7.41)	0.758	1.08 (0.18, 6.57)	0.936	0.90 (0.24, 3.33)	0.805	0.84 (0.22. 3.27)	0.799

^1^ Data from one infant consuming soy formula and two underweight infants at the follow-up visit were not included in the analysis. *Adjusted for age of the mother and infant, sex of the infant, mother’s level of education, and pregnancy weight gain status.

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
