# Peer review of "Amount, Preparation and Type of Formula Consumed and Its Association with Weight Gain in Infants Participating in the WIC Program in Hawaii and Puerto Rico"

_nutrients, 2019, doi:10.3390/nu11030695_

Round 1
Reviewer 1 Report
Authors reported the association between amount, preparation and type of formula consumed and weight gain in infants participating in the WIC Program of Hawaii and Puerto Rico. This study is well conducted and the topic is very pertinent. I have the following comments and would suggest minor revisions.
1. Page 2, line 50 - posterior risk of obesity - do authors refer to obesity later in life? Please clarify and revise accordingly.
2. Page 3, line 135 - A z score of <2 should be changed to < -2
3. General comments - Methods/Results: Does the food frequency questionnaire (FFQ) include the method of mixing in formula fed babies (both for powder and liquid conc.). Including this information may help to evaluate the association between mixing errors with slow/rapid gain of weight.
4. General comments - Methods/Results: Did authors explore the association between pre-pregnancy BMI and infant weight gain characteristics. Also report any association noted between pregnancy weight gain and weight changes in infants.
Author Response
Reviewer 1
Authors reported the association between amount, preparation and type of formula consumed and weight gain in infants participating in the WIC Program of Hawaii and Puerto Rico. This study is well conducted and the topic is very pertinent. I have the following comments and would suggest minor revisions.
Comment | Changes in response to comments |
1. Page 2, line 50 - posterior risk of obesity - do authors refer to obesity later in life? Please clarify and revise accordingly. | Yes, we meant obesity later in life. It was modified accordingly. |
2. Page 3, line 135 - A z score of <2 should be changed to < -2 | Revised, as suggested. |
3. General comments - Methods/Results: Does the food frequency questionnaire (FFQ) include the method of mixing in formula fed babies (both for powder and liquid conc.). Including this information may help to evaluate the association between mixing errors with slow/rapid gain of weight. | The FFQ included the method of mixing formula but we did not detect errors in the mixing, it was done according to instructions. |
4. General comments - Methods/Results: Did authors explore the association between pre-pregnancy BMI and infant weight gain characteristics. Also report any association noted between pregnancy weight gain and weight changes in infants. | Yes, this was already reported in another publication. Please see link to the reference: https://www-ncbi-nlm-nih-gov.ezproxy.fiu.edu/pubmed/30524746
|

Reviewer 2 Report
Manuscript ID: nutrients-438550
Amount, preparation and type of formula consumed and its association with weight gain in infants of Hawaii and Puerto Rico participants of the WIC Program
Authors: Rafael Graulau, Jinan Banna, Maribel Campos, Cheryl Gibby, and Cristina Palacios *
This is a secondary analysis of data collected in WIC clinics in Hawaii and Puerto Rico as part of a multi-site trial seeking to improve infant weight in low-income minorities. The aim of the present was to evaluate the association between A) type of feeding, B) amount of infant formula, C) type of infant formula, and D) formula preparation during the first two months of life and the risk of overweight and rapid weight gain four months later.
Overall, the research question is interesting, but there are some major issues within the perception of the exposure and the outcome as well as the analytical strategy.
Major comment 1:
The classification of infant formulas as 1) hydrolysate, 2) casein-based, or 3) soy-based isn’t meaningful. The authors state that casein-based formula was more common (line 161), but is this 100% casein-based (which will be quite a surprise in my opinion)? Furthermore, the authors seem to term whey-based formula as hydrolysate, but hydrolysate formula might also be casein-based (e.g. Nutramigen from Mead Johnson).
Major comment 2:
Lines 78-79, “Results from this study could contribute to the new recommendations being developed for children from birth to 2 years of age in the US.” What recommendations? And what about the WHO breastfeeding recommendation? This is not mentioned anywhere.
Major comment 3:
Authors use weight-for-height Z-score (WHZ) as one of their outcome and defines rapid weight gain as increase > 0.67 in WHZ. This is not correct, rapid weight gain is defined as an increase > 0.67 SD in weight-for-age Z-score (WAZ). I can’t figure out how the authors did calculate the Z-scores – using some of the packages made for statistical programs, including SPSS, or?
This relates to table 3 as well – why not investigating WAZ or BMI-for-age Z-score (BMIZ)? These two are more common within this field of research.
Major comment 4:
How did the authors chose which variables to include in the adjust analyses? For example, table 5 is adjusted for number of children, age of the infant, maternal educational level, and pregnancy weight gain. I acknowledge the possibility, that the latter three are confounding, but how is the number of children confounding the association between type of feeding and weight at follow-up?
Did they consider checking for interaction between their exposure and the child’s sex?
Major comment 5:
Lines 201-217, discussion on rapid weight gain, differences between formula-fed and breastfed infants. I suggest you look into the work by Dewey et al., for example, Dewey KG, Heinig MJ, Nommsen LA, Peerson JM, Lönnerdal B. Growth of breast-fed and formula-fed infants from 0 to 18 months: the DARLING Study. Pediatrics 1992 Jun;89(6 Pt 1):1035-1041.
Major comment 6:
Line 255, catch-up growth: I think you should look for another reference than reference 43, which relates to (and is published in) Korean infants.
Minor comment 1:
The authors have not complied with the author guidelines of the Nutrients journal. The affiliations are not set up after the template, i.e. authors’ names should not be listed in front of each affiliation. Furthermore, the abstract should not be divided into sections and headings should not be present.
Minor comment 2:
There are some typos and inconsistencies throughout the paper. For example:
· Line 26 “socio- demopgraphic” should be “sociodemographic”.
· Lines 33 and 34 “adjOR” and “adj OR” spelled in two different ways. Avoid the usage of abbreviations in the abstract.
· Line 38 “larger and longer studies”, longer studies is an odd phrase, I suggest something like “studies with later (or longer) follow-up.”
· Line 93, the abbreviation of food frequency questionnaire (“FFQ”) should be presented here. It isn’t presented until line 109.
· Line 157 and Table 1: Gender is not a biological term. I suggest “sex” instead.
· Table 1: “Pregnancy weight gain”, the classification of low, adequate, and excessive pregnancy weight gain has not presented before.
· Line 194: There should be an “In” in the beginning of the conclusion.
Minor comment 3:
Lines 47-52. I wonder how the authors can mention rapid weight gain without referring to the work of Rolland-Cachera et al.? For example, Rolland-Cachera MF. Rate of growth in early life: a predictor of later health? Adv Exp Med Biol 2005;569:35-39. But also Zheng M, Lamb KE, Grimes C, Laws R, Bolton K, Ong KK, Campbell K. Rapid weight gain during infancy and subsequent adiposity: a systematic review and meta-analysis of evidence. Obes Rev 2018 Mar;19(3):321-332.
Minor comment 4:
Line 54, breastfeeding and its protection against obesity. This is very debated, and I think the authors’ statement is too definitive. Otherwise add “compared to non-breastfeeding”.
Minor comment 5:
Line 135, reference 25: Is this reference correct? I can’t see how it adds to the calculation of WHZ?
Minor comment 6:
Table 1, Race/ethnicity: The numbers exceed the group total of 162 (and 100%).
Minor comment 7:
Line 267-271, discussion on longitudinal data and infant feeding practices. You may find the following useful: Azad et al., Infant feeding and weight gain: Separating breast milk from breastfeeding and formula from food. Pediatrics 2018 Oct;142(4).
Author Response
We thank the reviewer for her/his comments. We made several changes in the revised manuscript. These changes are described below.
Reviewer 2
This is a secondary analysis of data collected in WIC clinics in Hawaii and Puerto Rico as part of a multi-site trial seeking to improve infant weight in low-income minorities. The aim of the present was to evaluate the association between A) type of feeding, B) amount of infant formula, C) type of infant formula, and D) formula preparation during the first two months of life and the risk of overweight and rapid weight gain four months later.
Overall, the research question is interesting, but there are some major issues within the perception of the exposure and the outcome as well as the analytical strategy.
Comment | Changes in response to comments |
Major comment 1: The classification of infant formulas as 1) hydrolysate, 2) casein-based, or 3) soy-based isn’t meaningful. The authors state that casein-based formula was more common (line 161), but is this 100% casein-based (which will be quite a surprise in my opinion)? Furthermore, the authors seem to term whey-based formula as hydrolysate, but hydrolysate formula might also be casein-based (e.g. Nutramigen from Mead Johnson). | We agree with the reviewer. After extensive review of type of formulas, we changed the analysis to: - Formulas not hydrolysate. - Formulas partially or extensively hydrolysate.
We changed the results accordingly. |
Major comment 2: Lines 78-79, “Results from this study could contribute to the new recommendations being developed for children from birth to 2 years of age in the US.” What recommendations? And what about the WHO breastfeeding recommendation? This is not mentioned anywhere. | The Dietary Guidelines for Americans (DGA) are being prepared for infants from birth to 24 months. Currently, the DGAs are only for 2 years and older. However, due to lack of data, more studies among this group has been requested: https://www.cnpp.usda.gov/birthto24months
Although there are WHO recommendations on breastfeeding, and it is the preferred method for feeding infants, guidelines are needed for formula feeding as most US infants are not breastfed.
We added the following to the introduction:
4th paragraph: Breastfeeding is the feeding method recommended by the World Health Organization [17]. However, this is low in the US. Data from the WIC program, a nutrition supplementation program in the US with more than 1.93 million infants participating nationally in 2016, showed that only 30.9% were breastfed (12.9% exclusively breastfed and 18.0% partially breastfed) while 69.1% were formula fed [18].
Last paragraph: Results from this study could contribute to the new Dietary Guidelines for Americans being developed for children from birth to 2 years of age in the US [23]. Although breastfeeding is the recommended method, this is not the main feeding method used in the US and studies are needed to understand the contribution of formula feeding to obesity. |
Major comment 3: Authors use weight-for-height Z-score (WHZ) as one of their outcome and defines rapid weight gain as increase > 0.67 in WHZ. This is not correct, rapid weight gain is defined as an increase > 0.67 SD in weight-for-age Z-score (WAZ). I can’t figure out how the authors did calculate the Z-scores – using some of the packages made for statistical programs, including SPSS, or? This relates to table 3 as well – why not investigating WAZ or BMI-for-age Z-score (BMIZ)? These two are more common within this field of research. | As explained in the methods section, under Infant’s weight and length, we wrote: Rapid weight gain was defined as a change greater than 0.67 standard deviations (SD) in weight-for-length z-score between the first and second assessment [4]. Adequate weight gain was considered as a change in weight-for-length z-score between -0.67 to +0.67 SD, while a slow weight gain is that below -0.67 SD.
These z scores were calculated at each time point using the WHO AnthroPlus macro [30].
We used WHZ as this is the method used by WHO and several studies:
https://www.ncbi.nlm.nih.gov/pubmed/27231870 https://www.ncbi.nlm.nih.gov/pubmed/30524746 https://www.ncbi.nlm.nih.gov/pubmed/24204979 https://www.ncbi.nlm.nih.gov/pubmed/22434753 https://www.ncbi.nlm.nih.gov/pubmed/20863516 https://www.ncbi.nlm.nih.gov/pubmed/1594343 https://www.ncbi.nlm.nih.gov/pubmed/30249624
To clarify, we added the following in the revised manuscript.
Weight-for-length z scores were calculated at each time point using the WHO AnthroPlus macro [30]… |
Major comment 4: How did the authors chose which variables to include in the adjust analyses? For example, table 5 is adjusted for number of children, age of the infant, maternal educational level, and pregnancy weight gain. I acknowledge the possibility, that the latter three are confounding, but how is the number of children confounding the association between type of feeding and weight at follow-up? Did they consider checking for interaction between their exposure and the child’s sex? | As stated in the statistical analysis, we conducted bivariate analyses between the main outcome variables and socio-demographic variables to identify confounders. However, based on the reviewer’s suggestion, we also added sex of the baby. We also included age of the mother as another confounder, as it has also been shown to be an important determinant of infant obesity. Baby’s sex was not significant in any of the models. |
Major comment 5: Lines 201-217, discussion on rapid weight gain, differences between formula-fed and breastfed infants. I suggest you look into the work by Dewey et al., for example, Dewey KG, Heinig MJ, Nommsen LA, Peerson JM, Lönnerdal B. Growth of breast-fed and formula-fed infants from 0 to 18 months: the DARLING Study. Pediatrics 1992 Jun;89(6 Pt 1):1035-1041. | We added Dewey’s study (The Darling study in the discussion), as well as other references based on other comments made by the reviewer. See below the changes done in the discussion……
Most prospective studies have found a positive association between formula feeding with rapid weight gain [13–15,33–37], showing a different pattern of weight gain by type of feeding. For example, the Darling study following 87 infants (46 breastfed and 41 formula fed) for 12 months showed a similar WHL z scores between breastfed and formula fed infants during the first 4 months, but then from the 5th to 18th month, WHL z score was consistently higher among formula fed infants [35]. |
Major comment 6: Line 255, catch-up growth: I think you should look for another reference than reference 43, which relates to (and is published in) Korean infants. | This reference (Cho WK and Suh BK, Catch-up growth and catch-up fat in children born small for gestational age) was published as PMC in English. https://www.ncbi.nlm.nih.gov/pmc/articles/PMC4753194/ But following the suggestion from the reviewer, we searched for another reference and replaced this reference with a systematic review that seemed more appropriate. https://www.ncbi.nlm.nih.gov/pubmed/29691880
|
Minor comment 1: The authors have not complied with the author guidelines of the Nutrients journal. The affiliations are not set up after the template, i.e. authors’ names should not be listed in front of each affiliation. Furthermore, the abstract should not be divided into sections and headings should not be present. | The affiliations in the front page and the abstract were modified to comply with the journal’s guidelines. |
Minor comment 2: There are some typos and inconsistencies throughout the paper. For example: · Line 26 “socio- demopgraphic” should be “sociodemographic”. · Lines 33 and 34 “adjOR” and “adj OR” spelled in two different ways. Avoid the usage of abbreviations in the abstract. · Line 38 “larger and longer studies”, longer studies is an odd phrase, I suggest something like “studies with later (or longer) follow-up.” · Line 93, the abbreviation of food frequency questionnaire (“FFQ”) should be presented here. It isn’t presented until line 109. · Line 157 and Table 1: Gender is not a biological term. I suggest “sex” instead. · Table 1: “Pregnancy weight gain”, the classification of low, adequate, and excessive pregnancy weight gain has not presented before. · Line 194: There should be an “In” in the beginning of the conclusion. | We revised the document and corrected these errors.
For the comment about Table 1, we added the following in methods:
It also included a question about weight gain during pregnancy; this was classified as low, adequate or high based on the recommended weight gain by the Institute of Medicine [26]..
|
Minor comment 3: Lines 47-52. I wonder how the authors can mention rapid weight gain without referring to the work of Rolland-Cachera et al.? For example, Rolland-Cachera MF. Rate of growth in early life: a predictor of later health? Adv Exp Med Biol 2005;569:35-39. But also Zheng M, Lamb KE, Grimes C, Laws R, Bolton K, Ong KK, Campbell K. Rapid weight gain during infancy and subsequent adiposity: a systematic review and meta-analysis of evidence. Obes Rev 2018 Mar;19(3):321-332. | We included these reference in the introduction:
Growth is more rapid during early infancy than at any other time during the human life cycle, and studies have confirmed the association between rapid weight gain and risk of obesity later in life [4–7], particularly if the rapid weight gain occurs within the first months of life [8]. |
Minor comment 4: Line 54, breastfeeding and its protection against obesity. This is very debated, and I think the authors’ statement is too definitive. Otherwise add “compared to non-breastfeeding”. | We changed this to:
Breastfeeding has been long recognized to help reduce the risk of obesity compared to non-breastfeeding [10,11]. |
Minor comment 5: Line 135, reference 25: Is this reference correct? I can’t see how it adds to the calculation of WHZ? | Thanks for noticing this mistake, it was replaced with the reference from the WHO Child Growth Standards |
Minor comment 6: Table 1, Race/ethnicity: The numbers exceed the group total of 162 (and 100%). | In the US, Hispanics are considered an “ethnicity classification” not race. To clarify, we separated Ethnicity from Race in the table and the % now sum 100% in both categories. |
Minor comment 7: Line 267-271, discussion on longitudinal data and infant feeding practices. You may find the following useful: Azad et al., Infant feeding and weight gain: Separating breast milk from breastfeeding and formula from food. Pediatrics 2018 Oct;142(4). | We agree with the reviewer that this is an important reference. We included the following in the 2nd paragraph of the discussion:
The Canadian Healthy Infant Longitudinal Development (CHILD) birth cohort, which followed 2553 infants from birth up to 12 months of age also found that non-exclusive or partial breastfeeding for 3 or 6 months to be associated with rapid weight gain [37].
|
